# Ground-truth-free deep learning approach for accelerated quantitative parameter mapping with memory efficient learning

Naoto Fujita[1], Suguru Yokosawa[2], Toru Shirai[2], Yasuhiko Terada[1]*

**1** Institute of Pure and Applied Sciences, University of Tsukuba, Tsukuba, Japan, **2** FUJIFILM Corporation, Medical Systems Research and Development Center, Imaging Research Group, Minato City, Japan

* terada.yasuhiko.fu@u.tsukuba.ac.jp

## Abstract

Quantitative MRI (qMRI) requires the acquisition of multiple images with parameter changes, resulting in longer measurement times than conventional imaging. Deep learning (DL) for image reconstruction has shown a significant reduction in acquisition time and improved image quality. In qMRI, where the image contrast varies between sequences, preparing large, fully-sampled (FS) datasets is challenging. Recently, methods that do not require FS data such as self-supervised learning (SSL) and zero-shot self-supervised learning (ZSSSL) have been proposed. Another challenge is the large GPU memory requirement for DL-based qMRI image reconstruction, owing to the simultaneous processing of multiple contrast images. In this context, Kellman et al. proposed memory-efficient learning (MEL) to save the GPU memory. This study evaluated SSL and ZSSSL frameworks with MEL to accelerate qMRI. Three experiments were conducted using the following sequences: 2D T2 mapping/MSME (Experiment 1), 3D T1 mapping/VFA-SPGR (Experiment 2), and 3D T2 mapping/DESS (Experiment 3). Each experiment used the undersampled k-space data under acceleration factors of 4, 8, and 12. The reconstructed maps were evaluated using quantitative metrics. In this study, we performed three qMRI reconstruction measurements and compared the performance of the SL- and GT-free learning methods, SSL and ZSSSL. Overall, the performances of SSL and ZSSSL were only slightly inferior to those of SL, even under high AF conditions. The quantitative errors in diagnostically important tissues (WM, GM, and meniscus) were small, demonstrating that SL and ZSSSL performed comparably. Additionally, by incorporating a GPU memory-saving implementation, we demonstrated that the network can operate on a GPU with a small memory (<8GB) with minimal speed reduction. This study demonstrates the effectiveness of memory-efficient GT-free learning methods using MEL to accelerate qMRI.

**Data availability statement:** All datasets used in this study are publicly available from the Brainweb dataset (https://brainweb.bic.mni. mcgill.ca/brainweb/) and the SKM-TEA dataset (https://github.com/StanfordMIMI/skm-tea).

**Funding:** This work was supported by JSPS KAKENHI Grant Number JP24K00891 and JST BOOST, Japan Grant Number JPMJBS2414. The funder provided support in the form of salaries for authors SY and TS, but did not have any additional role in the study design, data collection and analysis, decision to publish, or preparation of the manuscript. The specific roles of these authors are articulated in the 'Author Contributions' section. There was no additional external funding received for this study.

**Competing interests:** SY and TS are employees of FUJIFILM Corporation. This does not alter our adherence to PLOS ONE policies on sharing data and materials. The authors have no other competing interests to declare.

## Introduction

Quantitative MRI (qMRI), which allows the direct imaging of quantitative MR parameter values, is expected to be useful in clinical research [1,2]. In contrast, quantitative images provide objective and standardized measurements of tissue parameters. In addition, quantification allows the integration and analysis of imaging data obtained from different scanners, institutions, and patients.

qMRI requires the acquisition of multiple images with changes in sequence parameters, resulting in longer measurement times than conventional imaging [3]. Therefore, acceleration techniques focused on qMRI have been actively developed [4–6], including the proposal of specific sequences for fast qMRI. Compressed sensing (CS) [7], which leverages the inherent sparsity of MR images and parallel imaging (PI) [8–11], which uses multiple receiver coils are also used to accelerate qMRI by reducing the number of data points acquired.

Recently, significant reductions in acquisition time and improvements in image quality have been reported when using deep learning (DL) for image reconstruction [12–15]. The DL framework commonly used in image reconstruction is supervised learning (SL), where many fully sampled (FS) data that satisfy the Nyquist condition are used to train the model parameters for reconstruction from undersampled (US) data that do not satisfy the Nyquist condition. Reconstruction models using SL have also been applied to accelerate qMRI.

Extending SL-based image reconstruction to qMRI is promising; however, two major challenges remain. First, SL methods require a large amount of FS data [12,13,16,17], which is time-consuming and labor-intensive. To address this problem, large datasets such as fastMRI [18] have been published. However, in qMRI, where the image contrast varies from sequence to sequence, acquiring large amounts of raw FS data is more challenging, and only a limited number of datasets are publicly available. Second, there are concerns about performance degradation in the SL framework when a domain shift occurs (i.e., when data acquisition conditions and imaging targets differ between training and inference) [19–21]. Thus, the data domain must be the same during training and inference to demonstrate high reconstruction performance. This domain shift issue is even more pronounced in qMRI, where SL reconstruction models tend to be specific for certain quantitative parameters and sequences.

To address these issues, methods that do not require FS data, such as self-supervised learning (SSL) [22] and zero-shot self-supervised learning (ZSSSL) [23], have been proposed for DL image reconstruction. In 2020, Yaman et al. proposed SSL using data undersampling (SSDU) [22] for single-contrast images. As for ZSSSL, Yaman et al. extended SSDU to zero-shot SSDU (ZS-SSDU) [23]. Interestingly, ZS-SSDU uses only US data for reconstruction and does not require additional training data. SSL and ZSSSL have also been applied to qMRI. For SSL, Liu et al. proposed reference-free latent map extraction (RELAX) [24], which combines the regularization loss of quantitative images with data consistency loss using US k-space data. For ZSSSL, Jun et al. [25] combined ZS-SSDU with subspace

reconstruction for 3D-QALAS, achieving a nine fold acceleration in T1, T2, and PD mapping. Bian et al. [26] extended the RELAX framework using an optimization algorithm that employs iterative model-based qMRI reconstruction in a DL framework.

Although several excellent SSL and ZSSSL models have been proposed, their effectiveness for qMRI has not yet been verified, and a performance evaluation against SL has not yet been conducted. Additionally, many of these network implementations are not designed to handle high-dimensional data in qMRI, making it difficult to apply them beyond a limited number of sequences.

Another practical challenge in DL-based qMRI image reconstruction is the requirement for a large amount of GPU memory. This problem is common to the SL and SSL reconstruction models. qMRI typically involves the handling of high-dimensional data because of the need to process multiple contrast images simultaneously. Thus, issues such as the inability to use networks for multicoil and multicontrast data owing to GPU memory limitations and the difficulty in handling large-matrix data have been highlighted. For example, RELAX-MORE requires substantial GPU memory (59 GB on a Tesla A100 instrument). To address this issue, Kellman et al. proposed memory-efficient learning (MEL) [27] to save GPU memory and then Wang et. al. demonstrated the feasibility of 3D and 2D dynamic MRI reconstruction [28]. MEL has not yet been applied to qMRI reconstruction models, including SSDU, and further validation is required.

To address these issues, this study aimed to evaluate the effectiveness of SSL and ZSSSL frameworks compared to SL frameworks in accelerating qMRI. Furthermore, we showed that MEL can reconstruct qMRI images on GPUs with limited memory. We extended SSDU to handle three qMRI measurements: variable flip angle spoiled gradient echo (VFA-SPGR), multiple slice multiple echo (MSME), and quantitative double echo steady-state (qDESS) sequences. The main contributions of this study are as follows.

To compare the performance of SL/SSL/ZSSSL in qMRI under the same conditions and to demonstrate the effectiveness and challenges of SSL and ZSSSL approaches compared to SL and extend the SSDU framework to multi-contrast image reconstruction problems and to provide implementation techniques to reduce GPU memory consumption.

## Background

### Problem formulation

In this section, we formulize the MRI sampling process. Under multi-coil conditions, the sampling process can be formalized as follows:

$$y_\Omega = E_\Omega x \tag{1}$$

where $x \in \mathbb{R}^{H \times W \times 2}$ is the target image, and $y_\Omega \in \mathbb{R}^{C \times H \times W \times 2}$ represents the US multi-coil k-space data. $\Omega$ is the sampling region, C is the number of receiving coils, and H×W is the image matrix size. The encoding operator $E_\omega : \mathbb{R}^{H \times W \times 2} \mapsto \mathbb{R}^{C \times H \times W \times 2}$ in PI is defined as:

$$E_\Omega = \begin{bmatrix} M_\Omega F C_1 \\ M_\Omega F C_2 \\ \vdots \\ M_\Omega F C_C \end{bmatrix} \tag{2}$$

where $C_i$ is the sensitivity of the receiving coil, $F$ is the discrete Fourier operator, and $M_\Omega$ is the sampling operator that fills unmeasured k-space points (outside $\Omega$) with zeros.

In this study, we extended the encoding operator in Equation ($2$) to the sampling process for multicontrast data. In multicoil/multicontrast data acquisition, different sampling regions are often employed for each contrast; thus, the multi-contrast extension of the encoding operator in Equation ($2$) is defined as:

$$E_\Omega = \begin{bmatrix} E_{\Omega_1} \\ E_{\Omega_2} \\ \vdots \\ E_{\Omega_P} \end{bmatrix}$$

(3)

Thus, the operator $E_\Omega : \mathbb{R}^{P \times C \times H \times W \times 2} \mapsto \mathbb{R}^{P \times H \times W \times 2}$ is the encoding operator for multi-contrast k-space data, where $P$ is the number of input contrast images, and $\Omega_i$ is the sampling region in k-space for the i-th contrast. Here, $\Omega = \{\Omega_1, \Omega_2, \ldots, \Omega_P\}$ represents the sampling regions for all contrasts. Using this operator, the sampling operation for multi-coil/ multi-contrast data can be expressed similarly to Equation ($1$). Henceforth, unless otherwise specified, the encoding operator in this paper refers to the multicontrast extended encoding operator defined in this section.

## Quantitative MRI sequences in this study

This section provides an overview of the qMRI sequences used in the present study. In this study, MSME, VFA-SPGR, and qDESS were adopted as standard methods for 2D T2, 3D T1, and fast 3D T2 mapping, respectively.

MSME is a pulse sequence used for T2 mapping that combines multi-slice imaging that excites multiple slices in parallel with the spin-echo technique, where a 90-degree RF pulse excites nuclear magnetization and a 180-degree RF pulse refocuses it to generate echo signals. MSME acquires multiple spin echoes at different echo times (TE) from a single excitation, thus enabling efficient T2 mapping. The signal model is expressed as follows:

$$S_i(I_0, T_2) = I_0 e^{-TE_i/T_2}$$

(4)

where $I_0$ and $T_2$ represent the proton density image and T2 map, respectively. $TE_i$ represents the echo time of the ith echo. Using the signal model in ($4$), the T2 map can be estimated using least-squares fitting (LSF).

VFA-SPGR is a widely used method for T1 mapping, particularly for acquiring 3D T1 maps [29]. This method uses SPGR sequences acquired multiple times with multiple FA and constant TR and TE. The signal model for the SPGR sequence is as follows:

$$S_i(I_0, T_1) = I_0 \frac{(1 - e^{-TR/T_1})}{1 - e^{-TR/T_1} \cos \alpha_i}$$

(5)

where $\alpha_i$ is the flip angle of the ith measurement. If there are at least two measurements with different flip angles, T1 mapping is possible using Equation ($5$).

qDESS is a 3D T2 mapping method based on the DESS sequence, a well-known, undistorted SNR-efficient 3D imaging technique [30]. In DESS, two contrast images, free induction decay (FID) signals, $S_{fid}$, and ECHO signals, $S_{echo}$, are obtained simultaneously per TR. T2 mapping is performed by utilizing the relationship between the FID signal and ECHO signal ratio as follows [31]:

$$\frac{S_{echo}}{S_{fid}} = e^{-2(TR-TE)/T_2}$$

(6)

In a study by Sveinsson et al. [32], T1 and diffusion effects were considered, particularly when focusing on the knee.

$$\frac{S_{echo}}{S_{fid}} = e^{-2(TR-TE)/T_2-\left(TR-\frac{\tau}{3}\right)\Delta k^2 D} \, \sin^2\left(\frac{\alpha}{2}\right)\left(\frac{1+e^{-TR/T_1-TR\Delta k^2 D}}{1-cos\alpha e^{-TR/T_1-TR\Delta k^2 D}}\right) \tag{7}$$

where TR and TE represent the repetition and echo time, respectively. $\alpha$ is the flip angle, D is the diffusivity, and the dephasing per unit length induced by the unbalanced gradient is denoted by $\Delta k = \gamma G \tau$, where G and τ are the spoiler amplitude and duration, respectively. In this study, following Sveinsson et al. [32], we focused on knee imaging and assumed T1 to be 1000 ms, with other sequence parameters based on the SKM-TEA dataset [33].

**Training strategy of deep learning MRI reconstruction**

In this section, the DL training method used in this study is described. In recent years, research on learning methods has been actively conducted with the development of DL networks. The datasets required differed depending on the learning method, and some datasets did not require a ground truth (GT). The representative learning methods include SL, SSL, and unsupervised learning (UL). In SL, a pair of model inputs and reference data are provided, and the error between the model output and reference data is evaluated using a loss function. The weights were updated to minimize the loss. In contrast, SSL assumes a scenario in which a GT is unavailable. In SSL, a pretext task, which is a task related to the original goal, is used for learning. Specifically, the synthetic GT generated from the input data is used as the teacher data, and by learning the transformation to this synthetic teacher data, the model learns the ability to transform the input data to the original GT. UL also assumes a situation in which the GT is unavailable; however, unlike SSL, it does not involve synthesizing teacher data. In the UL, a loss function that evaluates only the output data is used.

In addition to the learning methods, research has also focused on reducing the number of images required for learning. Methods with more than two learning samples are called few-shot learning, those with only one learning sample are called one-shot learning, and those with zero learning samples are called zero-shot (ZS) learning. The ZS learning targeted in this study uses only test data, meaning that there is no need to prepare the data for learning. Therefore, it is fundamentally impossible in SL frameworks that require GT; therefore, SSL or UL frameworks must be used.

SSDU is an SSL framework for image reconstruction proposed by Yaman et al. in 2020. In the SSL framework, no data are required and only US data can be used for training. In a typical SL framework, learning requires paired data from the US and FS k-space data. The learning process in SL is expressed as follows:

$$\widetilde{\theta} = \min_\theta \mathcal{L}\left(y_U, E_U f_{dnn}\left(y_\Omega, E_\Omega; \theta\right)\right) \tag{8}$$

where $\widetilde{\theta}$ is the parameter obtained through learning, L is the loss function, $f_{dnn}\left(y_\Omega, E_\Omega; \theta\right)$ is the DL network with $\theta$ as the parameter and $E_\Omega$ as the internal encoding operator, and $y_U$ is the FS data.

By contrast, in the SSDU framework, learning is performed without using the FS data. In this case, the original US region $\Omega$ is divided into two sampling regions: $\Theta$ and $\Lambda$. The learning process in SSL is expressed by the following equation.

$$\widetilde{\theta} = \min_\theta \mathcal{L}\left(y_\Lambda, E_\Lambda f_{dnn}\left(y_\Theta, E_\Theta; \theta\right)\right) \tag{9}$$

Here, $\Omega = \Theta \cap \Lambda$. The goal of the image reconstruction problem is to reconstruct FS k-space data from US k-space data, which can be rephrased as the problem of completing the unsampled region $\Omega$. In the SSL framework, the ability to properly complete the unsampled region $\Omega$ is evaluated by assessing whether the sampling region $\Lambda$, which is not included in the sampling region $\Theta$ of the input k-space data, can be appropriately reconstructed. Yaman et al. extended SSDU to a ZS learning application. In ZS learning, because only the test dataset is available, training and validation

datasets must be generated from the test dataset. Normally, different data are used for the training and validation datasets. However, they solved this problem using different sampling regions for each dataset. Specifically, the US region $\Omega$ is divided into three sampling regions $\Theta$, $\Lambda$, and $\Gamma$, so that $\Omega = \Theta \cap \Lambda \cap \Gamma$.

## Materials and methods

### Study workflow

Performance comparisons of the three learning methods (SL, SSL, and ZSSSL) were conducted using three qMRI sequences (MSME, VFA-SPGR, and DESS). L1-ESPILiT [34] was included in the comparison as the baseline reconstruction method. Fig 1 shows the workflow of this study. The FS dataset was synthesized or obtained from a public dataset, as described later. For each experiment, the dataset was divided into training, validation, and testing subsets. The US k-space data used for training and testing were generated by retrospective sampling from the FS data under different acceleration factors (AFs), which represent the degree of acceleration relative to the acquisition time of the FS data (e.g., AF = 2 corresponds to a two-fold acceleration).

### Network architecture

We used the architecture of an unrolled network. Unrolled networks are a class of DL methods that integrate iterative optimization algorithms with neural networks, combining physics-based modeling and data-driven learning. This approach is

**Fig 1. Workflow of this study.** (a) Dataset preparation procedure in this study. (b) Types of data used for training each network. (c) Performance evaluation of each network during testing.US: Undersampled, FS: Fully sampling, SL: Supervised Learning, ZSSSL: Zero-Shot Self-Supervised Learning, SSL: Self-Supervised Learning, SSIM: structural similarity, NRMSE: Normalized Root Mean Squared Error, GT: ground truth, AF: acceleration factor.

known as SoTA for image reconstruction [35]. In recent years, its high performance has led to its application in qMRI, and its effectiveness has been reported [17,25,26]. Therefore, in this study, we adopt an unrolled network as the backbone model.

We extend the network architecture proposed by Aggarwal et al. [14] for multicoil/multicontrast problems. The accelerated MR reconstruction problem under multicoil/multicontrast conditions is formulated as follows:

$$x_{rec} = \underset{x}{argmin} \ \|y_{\Omega} - E_{\Omega}x\|_2^2 + \lambda R(x),$$
(10)

where $R(x)$ is a specific regularization term, and $\lambda$ is a parameter that balances the data-consistency term and the regularization term. If $R(x)$ is an L2 norm or another typical convex regularization, iterative methods such as the CG method or gradient descent are often employed.

The input to the networks was the coil sensitivity and US multi-contrast k-space data, and the output was the reconstructed multi-contrast k-space data. To construct an unrolled network, similar to that of Aggawal et al. We adopted a DL-based L2 norm for the regularization term in Equation (10) as follows:

$$x_{rec} = \underset{x}{argmin} \ \|E_{\Omega}x - y_{\Omega}\|_2^2 + \lambda \|x - D_w(x; \theta)\|_2^2$$
(11)

Here, $D_w$ is called the denoiser, which is responsible for removing artifacts similar to noise from CS, and $\theta$ is the parameter of the convolutional neural network (CNN). Equation (11) can be transformed into a problem of alternating convex optimization between variables $x$ and $z$ by introducing a new intermediate variable $z_n$.

$$z_n = D_w(x_n; \theta)$$
(12)

$$x_{n+1} = \underset{x}{argmin} \ \|E_{\Omega}x - y_{\Omega}\|_2^2 + \lambda \|x - z_n\|_2^2$$
(13)

Equation (12) corresponds to the denoiser, and Equation (13) corresponds to the data consistency (DC) layer, representing the process of alternating optimization of variables $x$ and $z$. Equation (13) can be rewritten using the linear operator $Q = E_{\Omega}^H E_{\Omega} + \lambda I$ as follows:

$$x_{n+1} = \left(E_{\Omega}^H E_{\Omega} + \lambda I\right)^{-1} (E_{\Omega}^H y_{\Omega} + \lambda z_n) \Leftrightarrow x_{n+1} = Q^{-1}(E_{\Omega}^H y_{\Omega} + \lambda z_n)$$
(14)

As (14) is a linear inverse problem, it can be solved using the conjugate gradient method. The conjugate gradient method, employed by Aggarwal et al was used in this study.

The network architecture is shown in Fig 2. In this study, we employed a denoiser structure based on the residual network (ResNet) [36]. ResNet is a model that introduces residual connections and enables the effective training of deep networks. In our preliminary experiments, using a multilayer CNN without residual connections as a denoiser resulted in unstable training. Therefore, we adopted a ResNet-based architecture similar to that proposed by Yaman et al. ResNet consists of ten repeated residual blocks (RB), each containing a convolutional layer, ReLU activation, another convolutional layer, and a scaling layer. The input and output are connected through residual connections. The scaling layer multiplied the input values by a factor of α, which was introduced to stabilize learning. In this study, α was set to 0.1, the kernel size of the convolutional layers was 3×3, and the number of filters was 64. The number of update steps for the DC layer was set to ten.

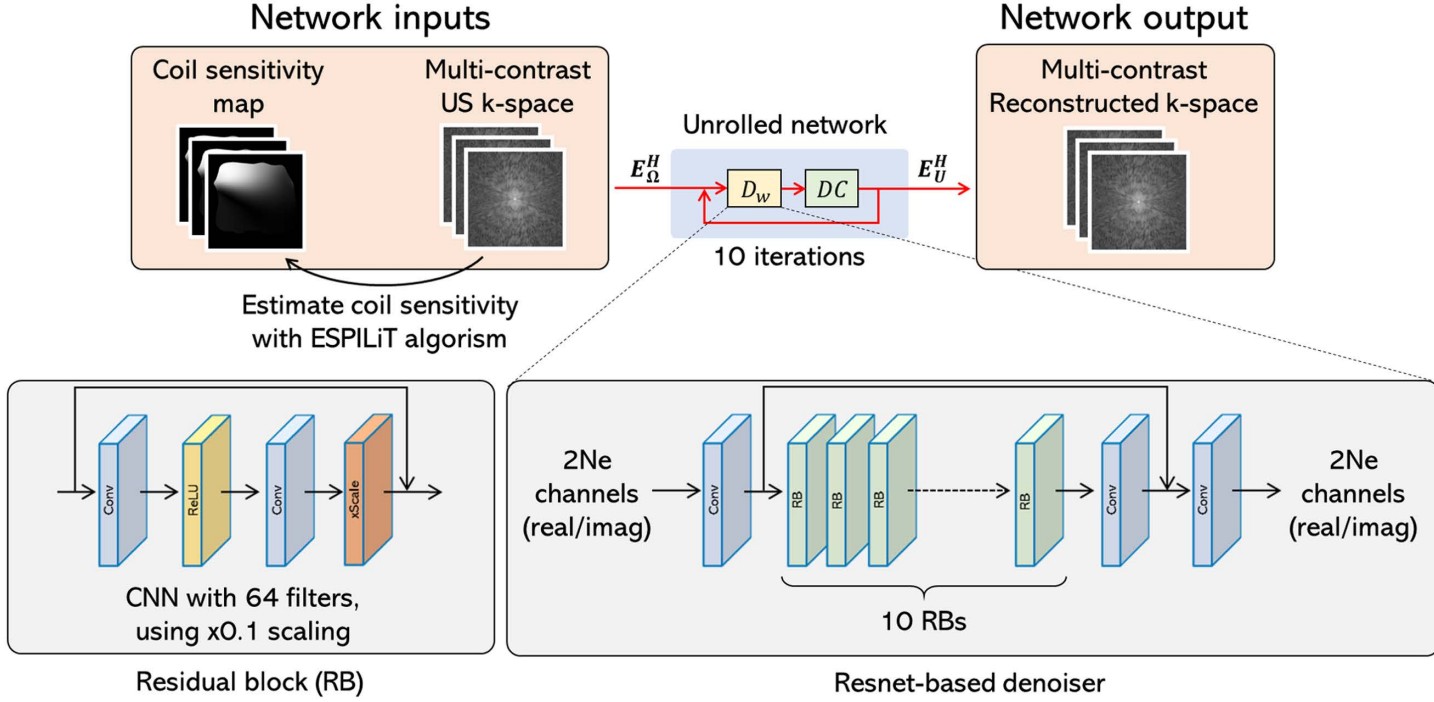

**Fig 2. Architecture of the image reconstruction model used in this study.** $D_w$ corresponds to denoising in Equation (12) and DC corresponds to data consistency in Equation (13). The multi-contrast US k-space is the input and the multi-contrast reconstruction k-space is the output. Because MRI images are complex-valued, the input and output of the denoising unit are concatenated in real and imaginary parts in the channel dimension direction. US: Undersampled, CNN: Convolutional neural network.

### Training strategy of this study

Three DL methods were compared: SL, SSL, and ZSSSL. The SSDU framework used in a previous study [23] was extended to the multicontrast problem because its implementation was publicly available and reproducible, and extending it to the multicontrast setting was relatively easy. These GT-free methods are highly useful in applications such as qMRI, where the cost of acquiring FS data is high. Fig 3 shows schematic diagrams of the multicontrast extensions of the SL and SSDU.

In each learning method, the training loss $\mathcal{L}_{train}$ was used to update the network parameters, and the validation loss $\mathcal{L}_{val}$ was used to optimize the weights. The losses under the SL condition are formulated as follows:

$$\mathcal{L}_{train} = \mathcal{L}_{val} = \mathcal{L}\left(y_U, E_U(f_{UO}(y_\Omega, E_\Omega; \theta))\right) \tag{15}$$

In the SSL condition, losses $\mathcal{L}_{train}$ and $\mathcal{L}_{val}$ are formulated as follows:

$$\mathcal{L}_{train} = \mathcal{L}_{val} = \mathcal{L}\left(y_\Lambda, E_\Lambda(f_{UO}(y_\Theta, E_\Theta; \theta))\right) \tag{16}$$

According to the original study, subsets $\Theta$ and $\Lambda$ were randomly selected such that $\Omega = \Theta \cap \Lambda$, with a 60% to 40% ratio for the split between $\Theta$ and $\Lambda$ similar to Yaman et al, which demonstrated strong performance in their study. In the ZSSSL condition, losses $\mathcal{L}_{train}$ and $\mathcal{L}_{val}$ are formulated as follows:

$$\mathcal{L}_{train} = \mathcal{L}\left(y_\Lambda, E_\Lambda(f_{UO}(y_\Theta, E_\Theta; \theta))\right) \tag{17}$$

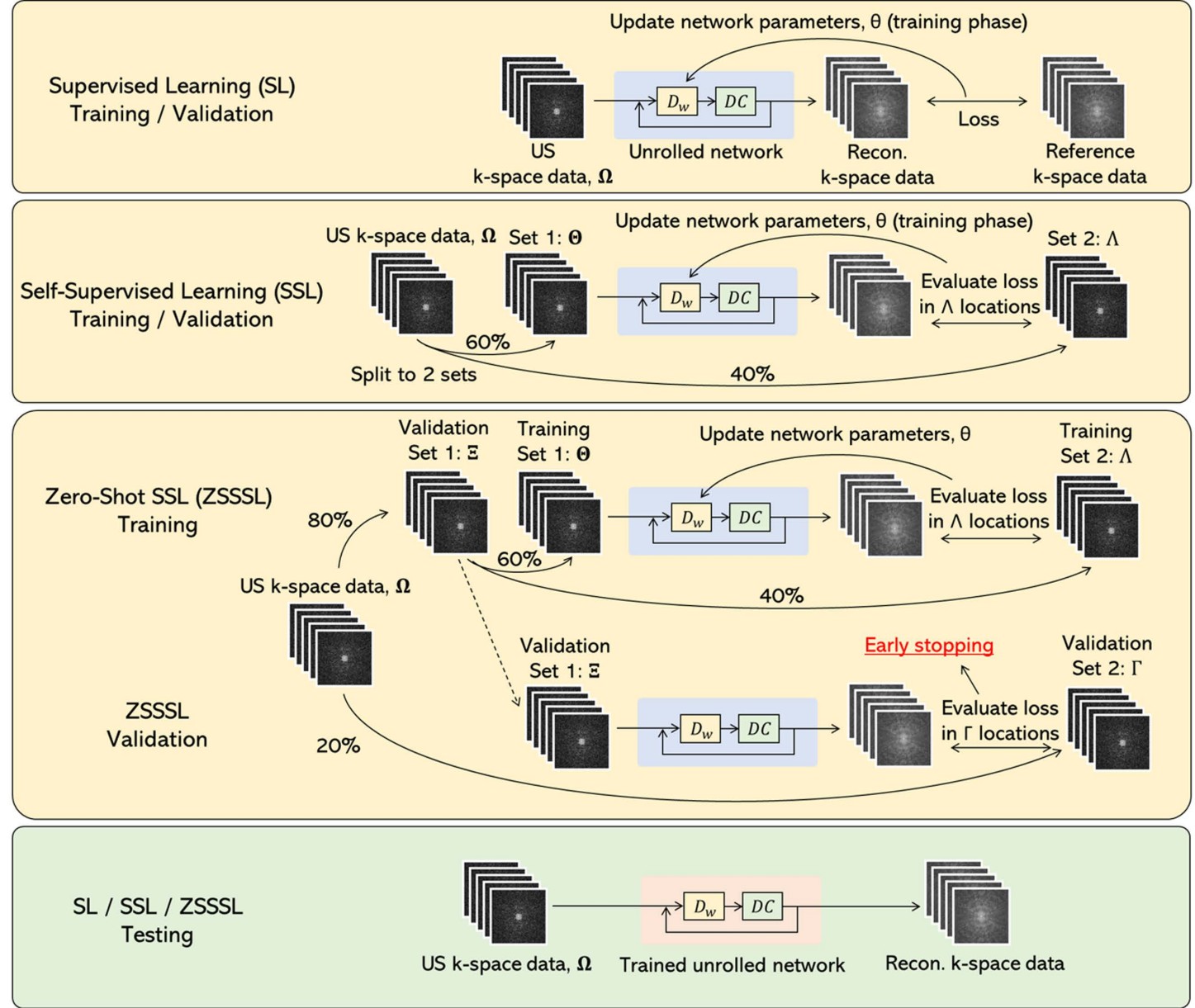

**Fig 3. Training strategy in this study.** The figure illustrates the training, validation, and testing procedures for supervised learning, self-supervised learning, and zero-shot SSL. $\Omega$ is the original US k-space. $\Theta$, $\Lambda$, $\Xi$, and $\Gamma$ are subsets of the k-spaces. US: Undersampled.

$$\mathcal{L}_{val} = \mathcal{L}\left(y_\Gamma, E_\Gamma\left(f_{UO}\left(y_\Xi, E_\Xi; \theta\right)\right)\right) \tag{18}$$

Subsets $\Theta$, $\Lambda$, $\Gamma$, and $\Xi$ were randomly selected such that $\Omega = \Xi \cap \Gamma$ and $\Xi = \Theta \cap \Lambda$, with an 80% to 20% ratio for the split between $\Xi$ and $\Gamma$, and a 60% to 40% ratio for the split between $\Theta$ and $\Lambda$ similar to Yaman et al. In ZSSSL, an early stopping strategy was employed, in which training was stopped if no improvement in $\mathcal{L}_{val}$ was observed for more than 25 epochs.

The loss function is a combination of the normalized L1 and L2 norms, defined as follows:

$$\mathcal{L}(u, v) = \frac{\|u - v\|_1}{\|u\|_1} + \frac{\|u - v\|_2}{\|u\|_2}$$

(19)

where $\mathcal{L}(u, v)$ is a loss function that takes $u$ and $v$ as inputs.

## Memory efficient learning strategy

We applied the MEL [27] strategy and gradient accumulation to conserve GPU memory for qMRI reconstruction. We implemented a gradient checkpoint strategy based on MEL [27]. Kellman et al. used a method in which, in addition to gradient checkpointing, the DC layer was recomputed in reverse to gain memory benefits. However, our observations indicated that this implementation did not yield significant GPU memory benefits, so we did not adopt it. Gradient checkpointing is a technique that conserves GPU memory by periodically saving "checkpoints" instead of storing all intermediate results during the forward pass and discarding intermediate outputs containing gradient information. During the backward pass, the necessary intermediate results are recomputed from the nearest checkpoint in the forward direction. In this study, we saved the intermediate outputs of denoiser $z_i$ as checkpoints. This process requires additional computation time to recompute the intermediate outputs of the denoiser during the backward pass. However, the maximum memory consumption is limited to the memory used to store the checkpoints and the memory required to retain the intermediate output of a denoiser computation step. Gradient accumulation saves memory by splitting the minibatch sent to the GPU and accumulating gradients, thus simultaneously reducing the amount of data sent to the GPU. In addition, we treated the 3D datasets as multiple k-space datasets by applying a Fourier transform to the readout direction of the k-space data. By combining these methods, we successfully employed the network in this study despite the high dimensionality of the target data.

## Experiments

The following experiments were conducted using three sequences: 2D T2 mapping/MSME (Experiment 1), 3D T1 mapping/VFA-SPGR (Experiment 2), and 3D T2 mapping/DESS (Experiment 3). These sequences have been commonly used in other qMRI studies [4,16,17,30,32,38]. Each experiment was conducted on the US k-space data under the AF conditions of 4 (AF4), 8 (AF8), and 12 (AF12). In single-contrast image reconstruction, AF4 is typically used, with AF8 considered a more challenging condition [18]. In this study, we also investigated AF12, anticipating further acceleration benefits from the use of multicontrast inputs. The datasets for Experiments 1 and 2 were created by simulating quantitative data from the BrainWeb database [37], while the dataset for Experiment 3 was created using a publicly available SKM-TEA dataset [33]. The sequence parameters and training conditions for each experiment are summarized in Table 1.

In the training phase, network training was performed based on each learning method. For SL, the network parameters were determined using training and validation datasets incorporating both FS and US data. In SSL, the network parameters were determined using only the US data in the training and validation datasets. In ZSSSL, only the US data from the test dataset were used.

To evaluate the performance of the trained models, ground-truth (GT) images for each contrast were synthesized from the FS data in the test dataset. Quantitative maps (proton density, T1, and T2) were generated using a signal model from the GT images for each contrast. US data from the test subset were input into the trained networks to generate reconstructed images for each contrast. Subsequently, quantitative value maps were calculated using the same procedure as that used for the GT.

I n this study, Python (version 3; Python Software Foundation, Wilmington, DE) was used, and all executions were performed inside a virtual container using the Docker Engine [38]. The base image used was nvcr.io/nvidia/pytorch:23.07-py3 (https://catalog.ngc.nvidia.com/orgs/nvidia/containers/pytorch). Each DL network was implemented using the PyTorch

**Table 1. Sequence parameters and training settings.**

| | Experiment 1 | Experiment 2 | Experiment 3 |
|---|---|---|---|
| Sequence name | multislice multiecho | spoiled gradient echo (SPGR) | double echo steady state (DESS) |
| Target tissue parameter | T2 | T1 | T2 |
| Experiment type | Simulation | Simulation | In-vivo |
| Flip angle (°) | 90 | 4/ 24 | 20 |
| Number of echoes | 16 | 2 | 2 |
| Number of coils | 6 | 6 | 8 |
| Echo time (ms) | 10 - 160 (Every 10) | 8 | 5.7/ 30.1 |
| Repetition time (ms) | 4000 | 18 | 17.9 |
| Slice thickness (mm) | 2 | – | – |
| Matrix size | 256×256 | 256×256 | 160×512 |
| Number of cases Training/ validation/ Test | 10/ 2/ 2 | 10/ 2/ 2 | 10/ 2/ 2 |
| Number of slices par case | 40 | 70 | 512 |
| learning rate (SL/SSL/ZSSSL) | $1.0 \times 10^{-4}$ / $1.0 \times 10^{-4}$ / $5.0 \times 10^{-4}$ | | |
| Number of epochs | 100* | | |
| Batchsize | 2** | | |

*We used an early stopping strategy in the ZSSSL; therefore, the number of epochs was not explicitly specified. **We used gradient accumulation owing to GPU memory limitations. SL: Supervised Learning, SSL: Self-Supervised Learning, ZSSSL: Zero-Shot Self-Supervised Learning.

framework. The hardware setup included a 64-bit Ubuntu Linux system (Canonical Ltd., London) with three GPUs: NVIDIA GeForce GTX 2080Ti (11 GB RAM), NVIDIA GeForce GTX 1080Ti (11 GB RAM), and NVIDIA GeForce GTX 3090 (24 GB RAM). The SSL/SL testing was conducted using a 64-bit Windows 11 system (Microsoft Ltd., Albuquerque, NM, USA) with an NVIDIA GeForce GTX 3080 (12 GB RAM).

## Simulation dataset (Experiments 1 and 2)

The simulation procedure is illustrated in Fig 4. We focused on multi-coil/multi-contrast data and simulated the signal intensity for each contrast and the coil sensitivity of the multi-coil system. Digital phantoms were obtained from the Brain-Web project (https://brainweb.bic.mni.mcgill.ca/), and 14 cases were simulated. In Experiment 1, the dataset was created using a Bloch simulation [39] based on the MSME sequence, and in Experiment 2, it was created from the SPGR signal model [40]. Next, six receiving coils were placed outside the field of view (FOV) region, and sensitivity maps were calculated using the Biot-Savart law. The coil placement is shown in Fig 5. The FS k-space data were obtained by multiplying the images generated from the digital phantoms by the calculated coil sensitivities. Gaussian noise with a standard deviation of 0.01% of the maximum k-space intensity was added to the created FS k-space. The sensitivity maps used for training and testing were regenerated from multicoil data using the ESPIRiT algorithm [41]. GT multi-contrast images were created from FS multi-contrast k-space data. The GT T1 map for Experiment 1 was created from GT multicontrast images using the SPGR signal model. The GT T2 map for Experiment 2 was created from the GT multicontrast images using the

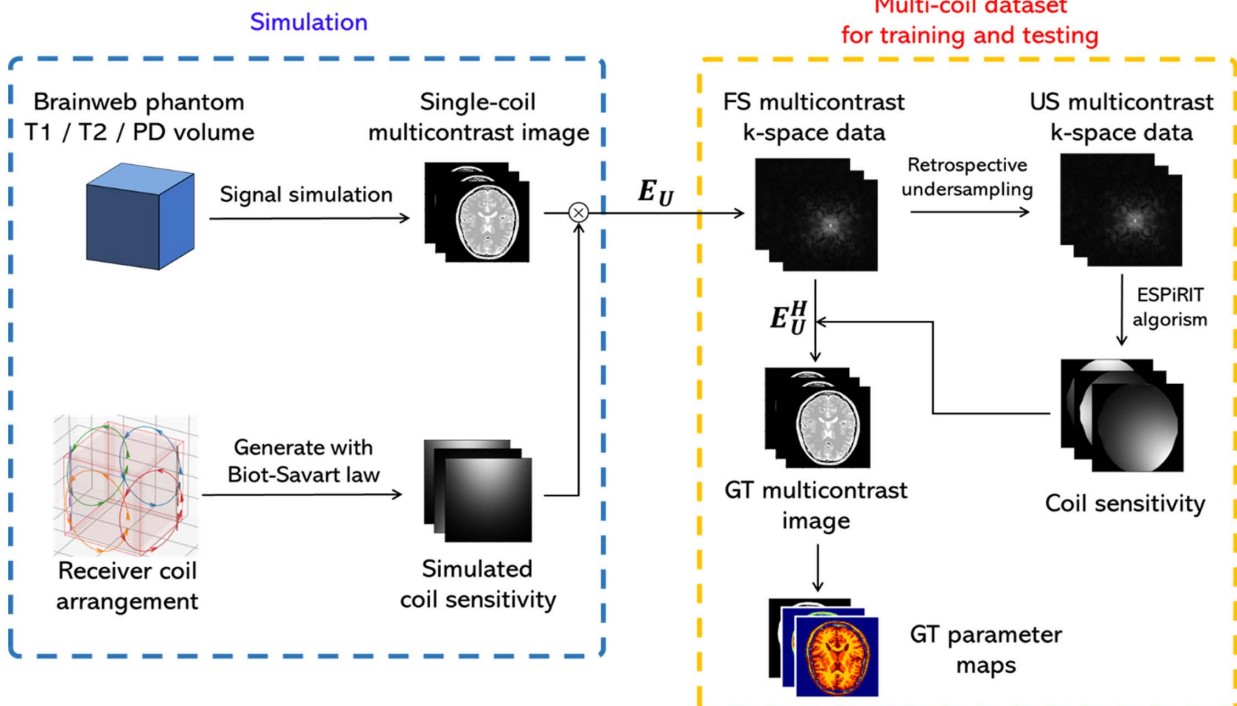

**Fig 4. Simulation procedures for Experiment 1 and Experiment 2.** The blue box represents the simulation process of multi-contrast data. The yellow box illustrates the procedure for generating the fully sampled (FS) and undersampled (US) k-space data and coil sensitivity maps for training and testing in this study. US: Undersampled, FS: Fully sampled, GT: ground truth.

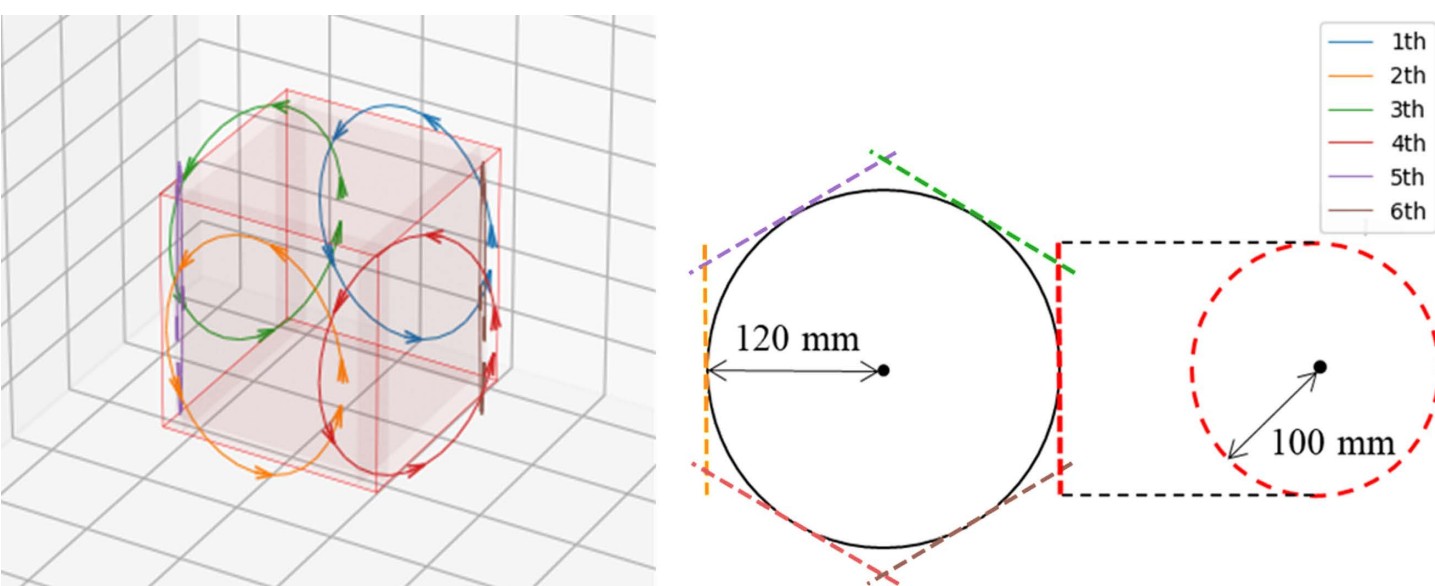

**Fig 5. coil arrangement used to simulate multi-coil signals.** (Left) The red cube represents the field of view (FOV), showing the spatial arrangement of the receiver coils. Different colors indicate the wiring patterns of each coil. (Right) A 2D projection of the coil arrangement. Each coil is modeled as a loop coil with a radius of 100 mm.

MSME signal model with LSF. The quantitative values from the multi-contrast image output by the network were calculated using the same procedure as that used for the GT.

### In-vivo dataset (Experiment 3)

We used the SKM-TEA dataset as the in vivo dataset. The SKM-TEA is an evaluation dataset for image reconstruction and segmentation tasks for knee T2 mapping using the DESS sequence. This dataset contains 3D volume data from 155 cases. Each volume includes 3D multicoil k-space data for both the FID and ECHO signals with different TEs as well as coil sensitivity maps estimated using the ESPIRiT algorithm. From this dataset, we searched for cases with 8-channel receiving coils and sequentially selected 14 cases. GT multi-contrast images were generated using the procedure described above. T2 maps were created from the GT multi-contrast images using the method described by Sveinsson et al. [30].

### Metrics and statistical analysis

The structural similarity (SSIM) was used as a metric to evaluate the errors in the reconstructed images for each contrast. The normalized root mean squared error (NRMSE) was used as a metric to evaluate the errors in the quantitative maps. The definitions of SSIM and NRMSE are provided in the S1 File. To prevent underestimation of NRMSE due to outliers, regions where the reconstructed quantitative maps had T1 values greater than 5000 ms or T2 values greater than 500 ms were excluded from the evaluation region $\Phi$. Additionally, regions where the GT signal intensity of the reconstructed contrast image was less than 5% of the maximum intensity were excluded because the estimation accuracy in these areas could not be sufficiently ensured. In Experiments 1 and 2, the evaluation region $\Phi$ was set to include the entire region and the specific region, including the white matter (WM) and gray matter (GM) regions with T1 < 1000 ms

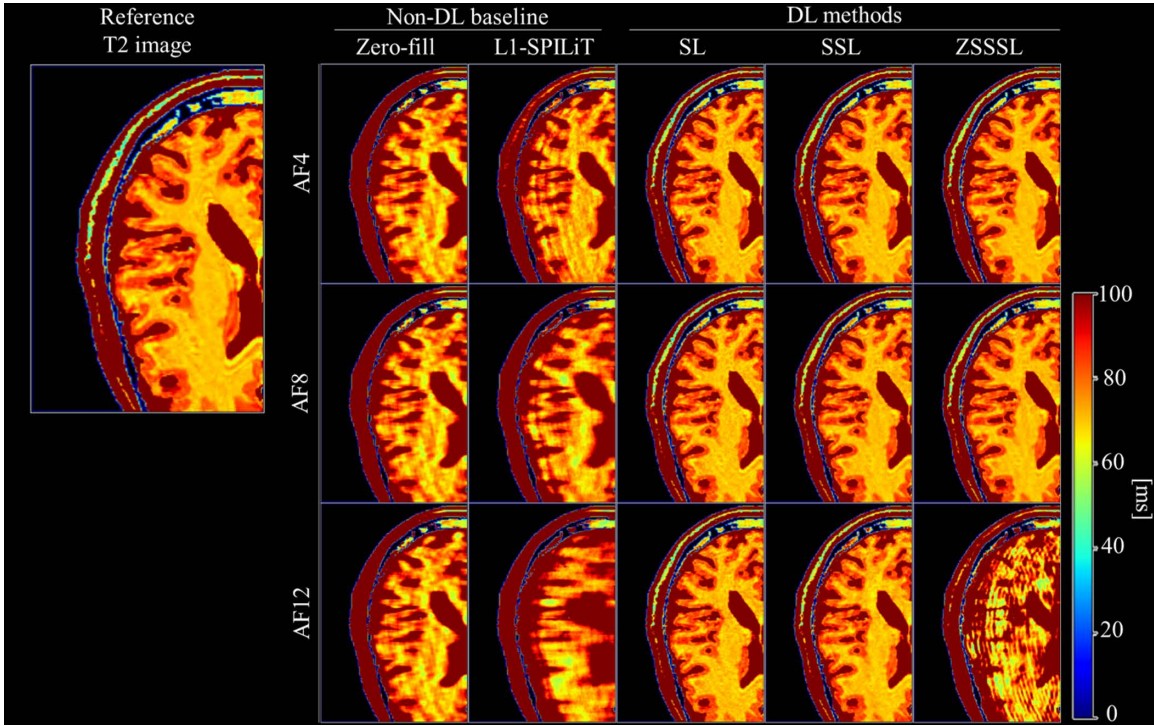

**Fig 6. Reconstructed images of Experiment 1.** (a) Reconstructed T2 images at AF4 - AF12. The unit of the colorbar is ms.SL: Supervised Learning, SSL: Self-Supervised Learning, ZSSSL: Zero-Shot Self-Supervised Learning.

and T2 < 120 ms. In Experiment 3, the evaluation region Φ was set to include only the meniscus region, as provided in the SKM-TEA dataset.

## Results

### Experiment 1

Fig 6 shows the reconstructed and reference T2 images from Experiment 1. S1–S3 Figs display the reconstructed contrast images with different TEs and AFs. The non-DL method, L1-ESPIRiT, performed significantly worse than the DL method for all AFs. In AF4 and AF8, there was little visual difference in the reconstructed T2 maps among the DL-based methods. However, for the reconstructed T2 images in AF12, noticeable performance degradation was observed for ZSSSL. These trends were also observed in the reconstructed contrast images (S1–S3 Figs).

Fig 7 shows the NRMSE values for the T2 maps. S1 Table in S1 Data displays the SSIM for the reconstructed contrast images with different TEs and AFs. The trends in the NRMSE of the T2 maps were consistent with those of the visual evaluation. The differences in the NRMSE between SL, SSL, and ZSSSL calculated for the entire image increased with the AF. SL and SSL showed NRMSEs of 0.23, 0.82, and 1.21 for SL, SSL, and ZSSSL, while ZSSSL showed 0.65, 0.82, and 20.37, respectively. Larger AF values resulted in larger differences between the SL method and the other methods. These trends were also observed in reconstructed contrast images (S1 Table in S1 Data). Among the GT-free methods, SSL and ZSSSL performed similarly, except for AF12, for which SSL performed better. In evaluating the GM and WM regions only, SSL and ZSSSL demonstrated comparable performance to SL.

### Experiment 2

Fig 8 shows the reconstructed and reference T1 images from experiment 2. S4–S6 Figs display the reconstructed contrast images with different FAs and AFs. As in Experiment 1, the DL-based methods exhibited fewer artifacts or blurring than L1-ESPIRiT. In addition, the significant performance degradation observed with ZSSSL in Experiment 1 was not observed in Experiment 2.

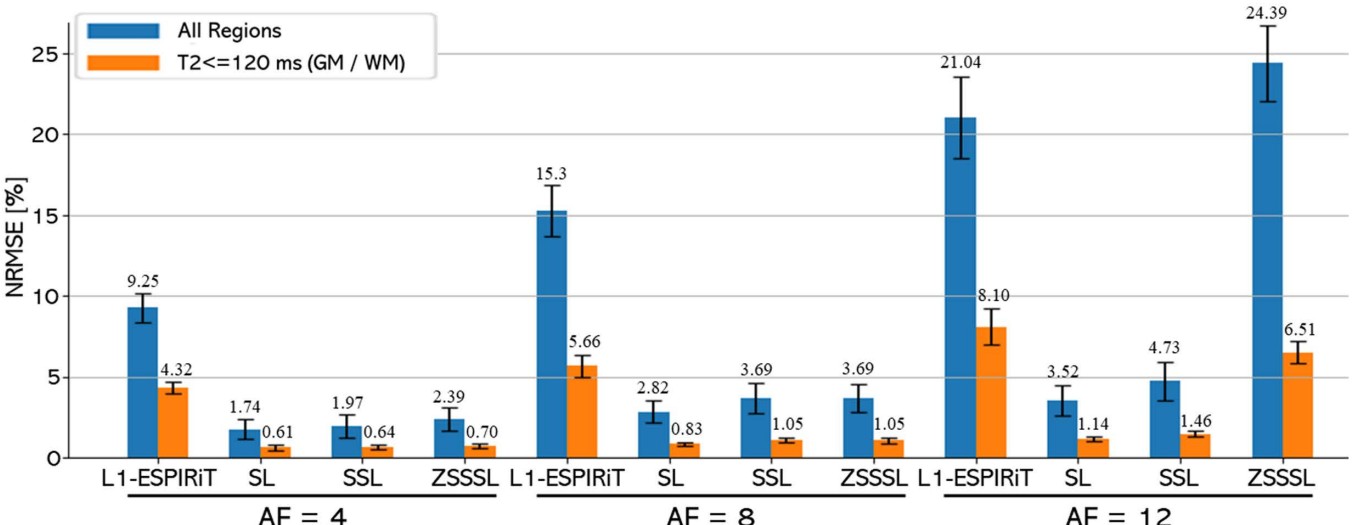

**Fig 7. Quantitative evaluation of Experiment 1.** Quantitative evaluation of the reconstructed T2 maps at AF4 - AF12. The evaluation was performed for both the entire region (blue) and the region with T2 ≤ 120 ms (orange), corresponding to Gray Matter (GM) and White Matter (WM). The numerical values above each bar indicate the mean NRMSE. SL: Supervised Learning, SSL: Self-Supervised Learning, ZSSSL: Zero-Shot Self-Supervised Learning.

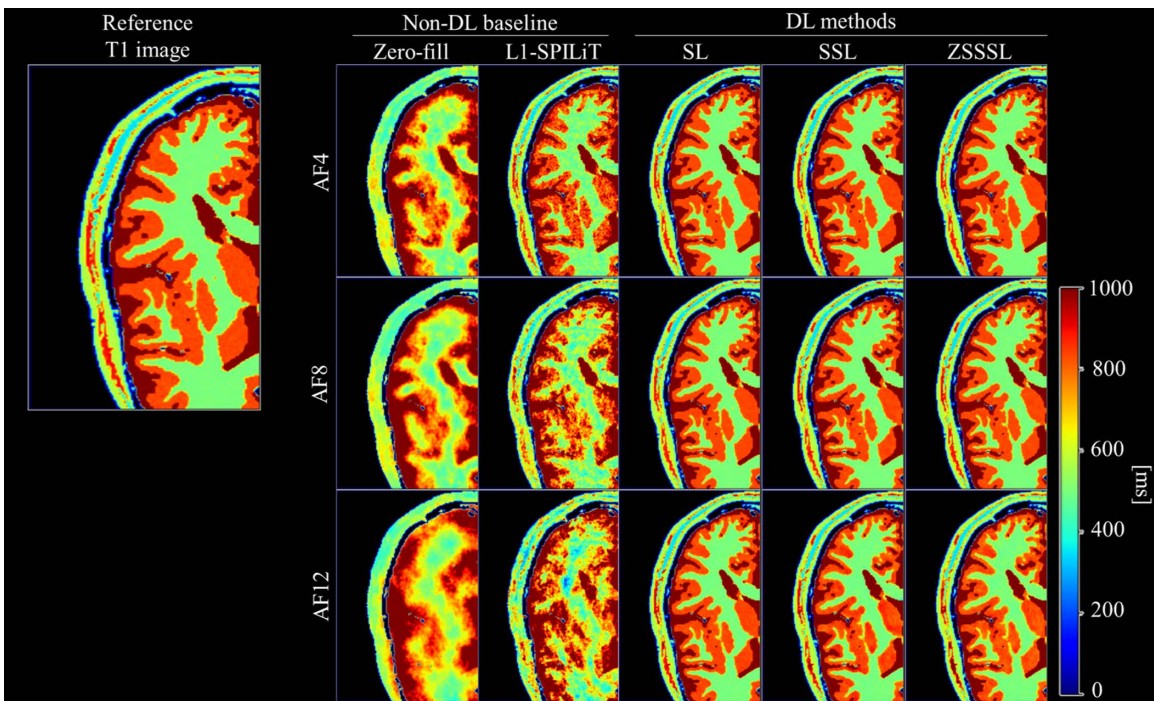

**Fig 8. Reconstructed images of Experiment 2.** (a) Reconstructed T1 images at AF4 - AF12. The unit of the colorbar is ms. SL: Supervised Learning, SSL: Self-Supervised Learning, ZSSSL: Zero-Shot Self-Supervised Learning.

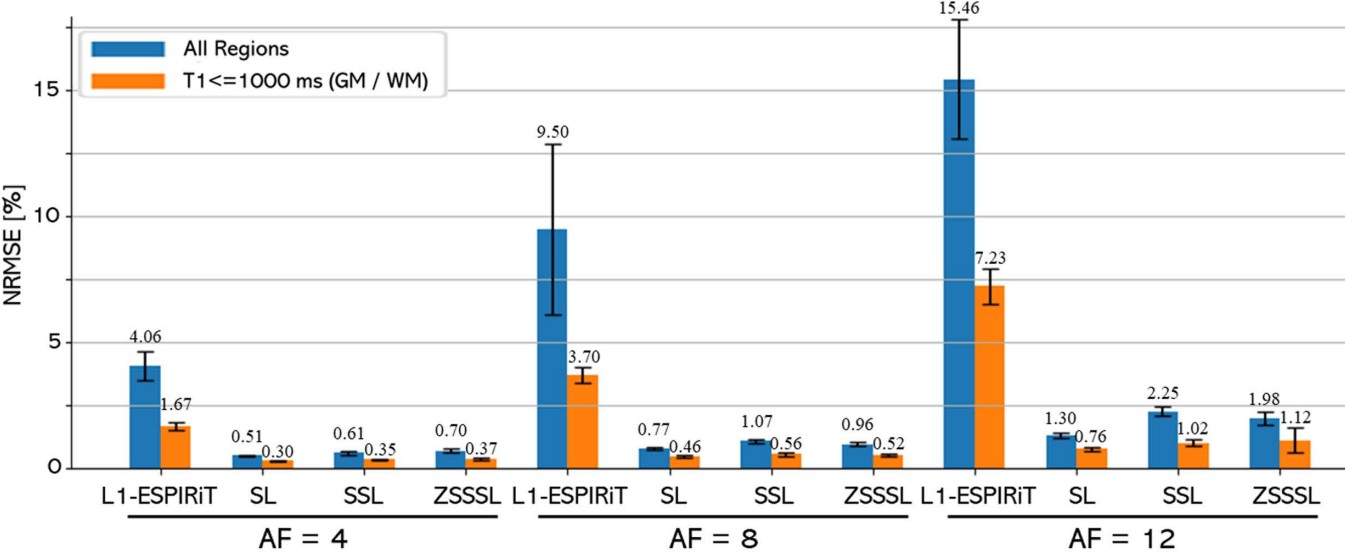

**Fig 9. Quantitative evaluation of Experiment 2.** Quantitative evaluation of the reconstructed T1 maps at AF4 - AF12. The evaluation was performed for both the entire region (blue) and the region with T1 ≤ 1000 ms (orange), corresponding to Gray Matter (GM) and White Matter (WM). The numerical values above each bar indicate the mean NRMSE. SL: Supervised Learning, SSL: Self-Supervised Learning, ZSSSL: Zero-Shot Self-Supervised Learning.

Fig 9 shows the NRMSE values for the T1 maps. S2 Table in S1 Data displays the SSIM for the reconstructed contrast images with different FAs and AFs. Similar to Experiment 1, the difference in the NRMSE between SL, SSL, and ZSSSL increased with increasing AF. Among the latter two methods, SSL outperformed ZSSSL at AF4, whereas ZSSSL outper-formed SSL at AF8 and AF12. The NRMSE for the T2 maps followed similar trends as in Experiment 1, and when the evaluation region was limited to the GM and WM, SSL and ZSSSL demonstrated performances comparable to that of SL.

## Experiment 3

Fig 10 shows the reconstructed and reference T2 images from Experiment 3, and magnified images of the cartilage. S7–S9 Figs display the reconstructed contrast images corresponding to the FID and ECHO signals. The DL-based methods accu-rately estimated the quantitative values of the meniscus region, which were not accurately estimated using L1-ESPIRiT (indicated by blue arrows). Small reconstruction errors were observed during the visual evaluation of the DL methods.

Fig 11 shows the NRMSE values for the T2 maps. S3 Table in S1 Data displays the SSIM of the reconstructed contrast images. In contrast to Experiments 1 and 2, the performance difference between SL and the other methods was within the range of the standard deviations, and the performance degradation was not large.

## Memory effectivity

The memory consumption results in Fig 12a show that MEL effectively reduced memory consumption, particularly when the number of layers was large. Fig 12b shows that the training speed was almost the same with and without the use of the MEL. The results in Fig 12a indicate that MEL significantly reduces memory consumption, particularly as the

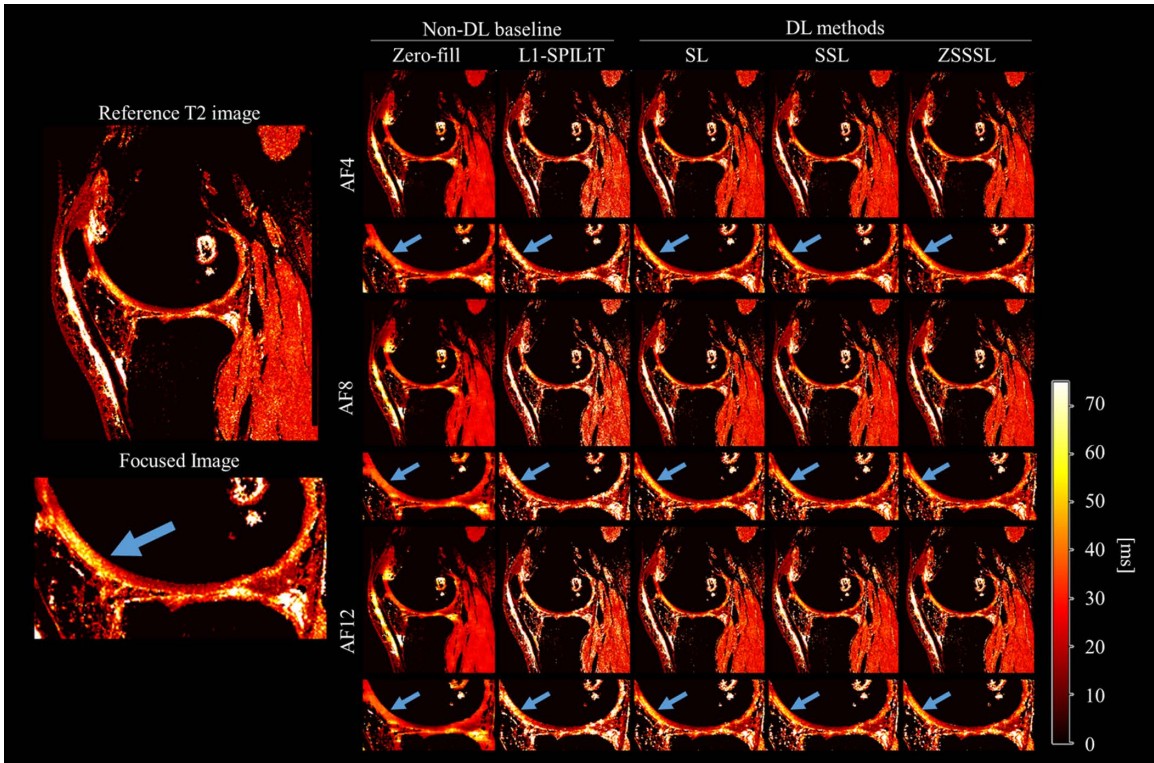

**Fig 10. Reconstructed images of Experiment 3.** (a) Reconstructed T2 images at AF4 - AF12. The unit of the colorbar is ms.SL: Supervised Learning, SSL: Self-Supervised Learning, ZSSSL: Zero-Shot Self-Supervised Learning.

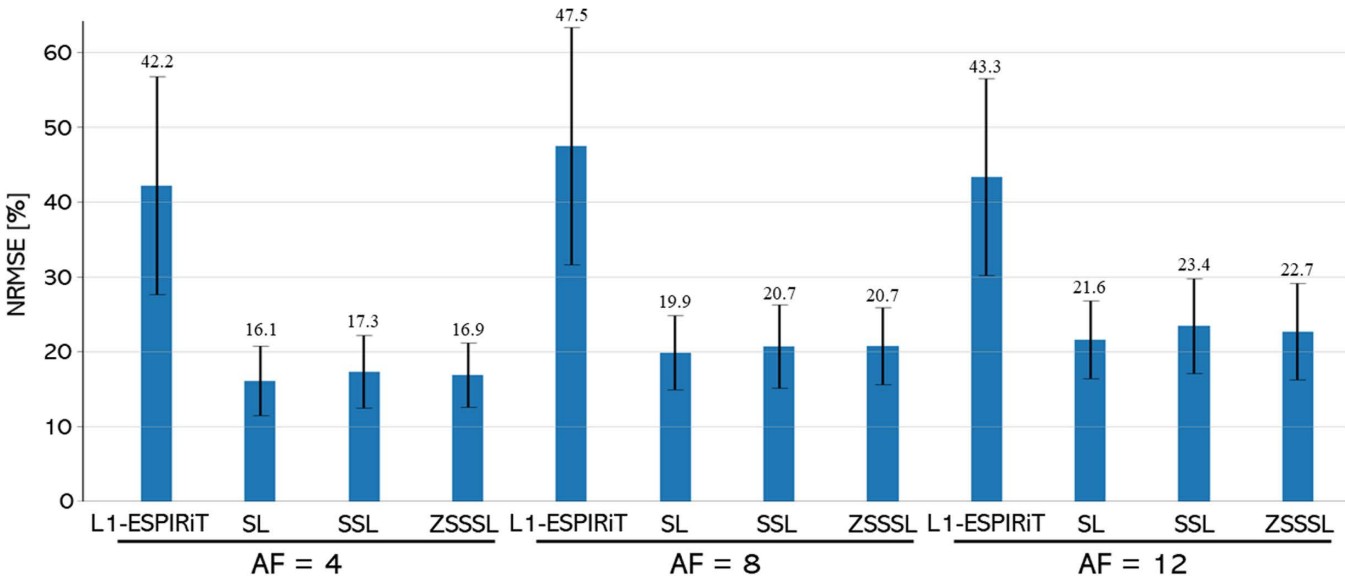

**Fig 11. Quantitative evaluation of Experiment 3.** Quantitative evaluation of the reconstructed T2 maps at AF4 - AF12. The numerical values above each bar indicate the mean NRMSE. SL: Supervised Learning, SSL: Self-Supervised Learning, ZSSSL: Zero-Shot Self-Supervised Learning.

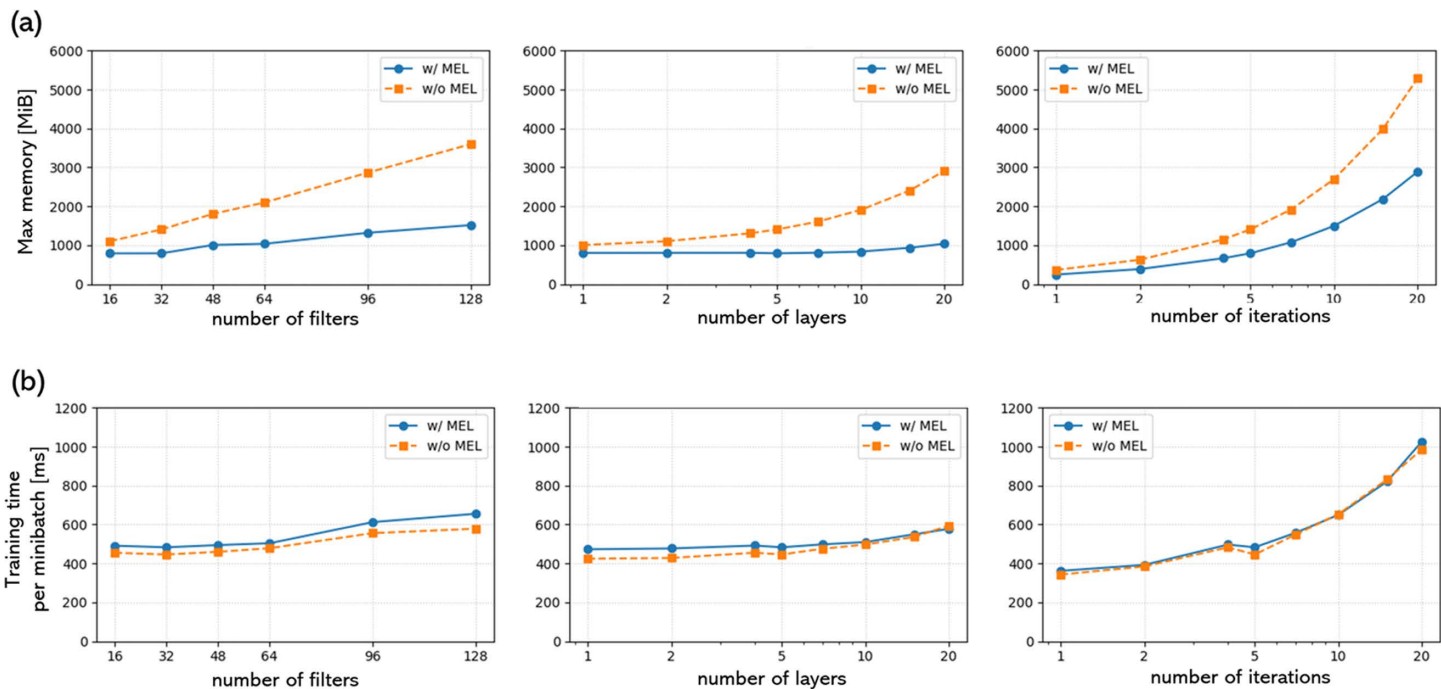

**Fig 12. Memory consumption and training time per minibatch in each network parameter settings.** (a) Maximum memory usage as a function of the number of filters, layers, and iterations. The solid blue line represents results with memory-efficient learning (w/ MEL), while the dashed orange line represents results without memory-efficient learning (w/o MEL). (b) Training time per minibatch.

number of layers, filters, and iterations increases. Without MEL, memory usage increases rapidly with network complexity, whereas MEL effectively suppresses this growth. By contrast, Fig 12b demonstrates that the training speed remains nearly unchanged regardless of the MEL usage, suggesting that the memory-efficient strategy does not introduce a significant computational overhead. In this study, the memory consumption was 7723 MB for Experiment 1, 1429 MB for Experiment 2, and 2000 MB for Experiment 3.

## Discussion

In this study, we performed three qMRI reconstruction measurements and compared the performance of the SL- and GT-free learning methods, SSL and ZSSSL. Overall, the performance of SSL and ZSSSL was only slightly inferior to that of SL, even under high-AF conditions. The quantitative errors in diagnostically important tissues (WM, GM, and meniscus) were small, demonstrating that SL and ZSSSL performed comparably. Additionally, by incorporating a GPU memory-saving implementation, we demonstrated that the network can operate on a GPU with a small memory (<8GB) with minimal speed reduction.

The GT-free learning method and memory-efficient strategy used in this study are expected to shorten the total examination time of protocols incorporating conventional qMRI and enable the integration of qMRI sequences that were previously challenging owing to long scan durations. The SSL technology allows the construction of DL models using only US data, even for qMRI sequences, where acquiring fully sampled (FS) data is time-consuming. Additionally, ZSSSL facilitates the application of DL models to emerging qMRI sequences that have not yet been widely adopted and lack large-scale training datasets. When incorporated into clinical workflows, these methods are expected to enhance objective diagnoses by utilizing qMRI as an imaging biomarker. However, to ensure the clinical relevance of the observed quantitative values, a standardized workflow from image acquisition to quantitative analysis, including diagnosis and treatment planning, must be established. Such standardization is essential alongside advancements in accelerated qMRI techniques.

A common trend across all the experiments was that the performance difference between SL and SSL increased as the AF increased. In SSL and ZSSSL, the US k-space data used during training are further split into data used for loss calculation and data for network input, and fewer data points are used for training. For example, in the case of AF4, the effective AFs for the network input were 6.6 (60% of the US k-space data) for SSL and 8.3 (48% of the US k-space data) for ZSSSL. For AF8, the difference in the effective AFs increased, and they were 13.2 for SSL and 16.6 for ZSSSL. Fewer sampling points in ZSSSL would lead to performance degradation for ZSSSL compared to SSL, especially at high AFs. This trend was also observed for AF12 in Experiment 1.

Although ZSSSL is highly effective because it does not require a training dataset, it has the drawback of lengthy reconstruction time. In this study, the reconstruction time per volume exceeded three hours, which may be acceptable in preclinical settings but is not feasible for clinical use. One potential solution to reduce the reconstruction time is to employ transfer learning (TL) using weights pretrained with SSL. Yaman et al. [23] reported that this TL approach effectively improves performance and reduces reconstruction time to less than one-tenth. Alternatively, TL using weights pretrained with natural images [42,43] or simulation-based training datasets [44] may also be effective. Another approach is to decrease the number of model parameters, which can be achieved through Neural Architecture Search (NAS) [45]. NAS is an automated technique for finding optimal neural network architectures by exploring a predefined search space, potentially reducing the model size while maintaining or improving the performance. Because the network used in this study has not yet undergone sufficient tuning of the hyperparameters such as the number of filters and layers, optimizing the model architecture remains a topic for future work.

This study is the first to implement MEL and gradient accumulation in qMRI reconstruction models to reduce memory consumption. In particular, MEL can estimate learning speed with little change while reducing memory consumption. In particular, unlike learning methods such as SL and SSL, which require pretraining, ZSSSL requires the gradient of the network weight to be stored in memory for backpropagation, even during inference. Increased memory consumption

increases the GPU hardware requirements for image reconstruction. The GPU memory consumption of ZSSSL for qMRI reconstruction in this study was maintained below 8GB, which is within the range of commonly available consumer GPUs.

In previous studies, Yaman et al. and Liu et al. applied SSL and ZSSSL to qMRI but only validated a single-coil scenario because of the lack of GPU memory. Bian et al. [26] proposed an unrolled network-type RELAX that requires a large memory. In particular, RELAX-MORE required up to 59 GB of memory for training. We expect similar memory reduction effects for these two qMRI applications using unrolled networks.

This study has two limitations. First, we restricted the GT-free SSL/ZSSSL model to SSDU, an SSL algorithm that adopts a k-space data-partitioning strategy. This is because SSDU is commonly used in practice [46–49]. However, other SSL algorithms [50–52] have also been proposed, and their performance when extended to qMRI should be evaluated in the future.

Second, the design of this study was retrospective, and its performance has not been verified using prospective data. In actual scans, imperfect imaging conditions, such as inaccurate sampling patterns due to eddy current effects, may degrade the reconstruction performance. For example, the B0 magnetic field inhomogeneity and eddy currents can cause geometric distortion and signal loss [53]. Therefore, when planning a prospective study, it may be necessary, for example to assess the impact of eddy currents on image distortion and trajectory deviation through spatiotemporal measurement of k-space trajectories using a field camera [54], and to evaluate the distortion itself through phantom study [55]. Validation with prospectively acquired US k-space data is a future issue, particularly in a study design such as the present study, which used only US k-space data.

## Conclusion

In this study, we evaluated the effectiveness of SSL and ZSSSL compared to SL in addressing the high-speed imaging problem in qMRI using three sequences: VFA-SPGR, MSME, and qDESS. The results showed that while SSL and ZSSSL tended to be slightly inferior to SL under high-AF conditions, overall, the quantitative error in diagnostically important tissues (WM, GM, and meniscus) was sufficiently small, demonstrating performance comparable to that of SL. These results indicate that GT-free learning methods such as SSL and ZSSSL can be applied to various qMRI sequences.

## Supporting information

**S1 Fig. Reconstructed contrast images in experiment 1 (AF4).** Error images relative to the fully sampled reference image are displayed with x10 scaling. SL: Supervised Learning, SSL: Self-Supervised Learning, ZSSSL: Zero-Shot Self-Supervised Learning.
(TIF)

**S2 Fig. Reconstructed contrast images in experiment 1 (AF8).** Error images relative to the fully sampled reference image are displayed with x10 scaling. SL: Supervised Learning, SSL: Self-Supervised Learning, ZSSSL: Zero-Shot Self-Supervised Learning.
(TIF)

**S3 Fig. Reconstructed contrast images in experiment 1 (AF12).** Error images relative to the fully sampled reference image are displayed with x10 scaling. SL: Supervised Learning, SSL: Self-Supervised Learning, ZSSSL: Zero-Shot Self-Supervised Learning.
(TIF)

**S4 Fig. Reconstructed contrast images in experiment 2 (AF4).** Error images relative to the fully sampled reference image are displayed with x10 scaling. SL: Supervised Learning, SSL: Self-Supervised Learning, ZSSSL: Zero-Shot Self-Supervised Learning.
(TIF)

**S5 Fig. Reconstructed contrast images in experiment 2 (AF8).** Error images relative to the fully sampled reference image are displayed with x10 scaling. SL: Supervised Learning, SSL: Self-Supervised Learning, ZSSSL: Zero-Shot Self-Supervised Learning.
(TIF)

**S6 Fig. Reconstructed contrast images in experiment 2 (AF12).** Error images relative to the fully sampled reference image are displayed with x10 scaling. SL: Supervised Learning, SSL: Self-Supervised Learning, ZSSSL: Zero-Shot Self-Supervised Learning.
(TIF)

**S7 Fig. Reconstructed Contrast images in experiment 3 (AF4).** Error images relative to the fully sampled reference image are displayed with x10 scaling. SL: Supervised Learning, SSL: Self-Supervised Learning, ZSSSL: Zero-Shot Self-Supervised Learning.
(TIF)

**S8 Fig. Reconstructed Contrast images in experiment 3 (AF8).** Error images relative to the fully sampled reference image are displayed with x10 scaling. SL: Supervised Learning, SSL: Self-Supervised Learning, ZSSSL: Zero-Shot Self-Supervised Learning.
(TIF)

**S9 Fig. Reconstructed Contrast images in experiment 3 (AF12).** Error images relative to the fully sampled reference image are displayed with x10 scaling. SL: Supervised Learning, SSL: Self-Supervised Learning, ZSSSL: Zero-Shot Self-Supervised Learning.
(TIF)

**S1 Data. S1 Table. Quantitative evaluation for contrast images in Experiment 1.** The numbers in the table represent SSIM. SL: Supervised Learning, SSL: Self-Supervised Learning, ZSSSL: Zero-Shot Self-Supervised Learning. **S2 Table. Quantitative evaluation for contrast images in Experiment 2.** The numbers in the table represent SSIM. SL: Supervised Learning, SSL: Self-Supervised Learning, ZSSSL: Zero-Shot Self-Supervised Learning. **S3 Table. Quantitative evaluation for contrast images in Experiment 3.** The numbers in the table represent SSIM. SL: Supervised Learning, SSL: Self-Supervised Learning, ZSSSL: Zero-Shot Self-Supervised Learning.
(PDF)

**S1 File. Performance metrics details.**
(DOCX)

## Author contributions

**Conceptualization:** Suguru Yokosawa, Toru Shirai, Yasuhiko Terada.

**Data curation:** Naoto Fujita, Yasuhiko Terada.

**Funding acquisition:** Yasuhiko Terada.

**Investigation:** Naoto Fujita, Yasuhiko Terada.

**Methodology:** Naoto Fujita, Suguru Yokosawa, Yasuhiko Terada.

**Project administration:** Suguru Yokosawa, Toru Shirai, Yasuhiko Terada.

**Software:** Naoto Fujita.

**Supervision:** Yasuhiko Terada.

**Validation:** Naoto Fujita.

**Visualization:** Naoto Fujita.

**Writing – original draft:** Naoto Fujita.

**Writing – review & editing:** Suguru Yokosawa, Toru Shirai, Yasuhiko Terada.

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
