## [Decision Letter · Decision Letter 0]

20 Feb 2025

PONE-D-24-54559Ground-truth-free deep learning approach for accelerated quantitative parameter mapping with memory efficient learningPLOS ONE

Dear Dr. Terada,

Thank you for submitting your manuscript to PLOS ONE. After careful consideration, we feel that it has merit but does not fully meet PLOS ONE’s publication criteria as it currently stands. Therefore, we invite you to submit a revised version of the manuscript that addresses the points raised during the review process.

We look forward to receiving your revised manuscript.

Kind regards,

Ibrahim Sadek, Ph.D.

Academic Editor

PLOS ONE

**Journal Requirements:**

1. When submitting your revision, we need you to address these additional requirements. Please ensure that your manuscript meets PLOS ONE's style requirements, including those for file naming. The PLOS ONE style templates can be found at https://journals.plos.org/plosone/s/file?id=wjVg/PLOSOne_formatting_sample_main_body.pdf and https://journals.plos.org/plosone/s/file?id=ba62/PLOSOne_formatting_sample_title_authors_affiliations.pdf 2. Please note that PLOS ONE has specific guidelines on code sharing for submissions in which author-generated code underpins the findings in the manuscript. In these cases, we expect all author-generated code to be made available without restrictions upon publication of the work. Please review our guidelines at https://journals.plos.org/plosone/s/materials-and-software-sharing#loc-sharing-code and ensure that your code is shared in a way that follows best practice and facilitates reproducibility and reuse. 3. Thank you for stating in your Funding Statement: This work was supported by JSPS KAKENHI Grant Number JP24K00891. This work was supported by JST BOOST, Japan Grant Number JPMJBS2414.  This work was supported by a grant from FUJIFILM Corporation.The funders had no role in study design, data collection and analysis, decision to publish, or preparation of the manuscript. Please provide an amended statement that declares *all* the funding or sources of support (whether external or internal to your organization) received during this study, as detailed online in our guide for authors at http://journals.plos.org/plosone/s/submit-now.  Please also include the statement “There was no additional external funding received for this study.” in your updated Funding Statement. Please include your amended Funding Statement within your cover letter. We will change the online submission form on your behalf. 4. Thank you for stating the following in the Acknowledgments Section of your manuscript: This work was supported by JSPS KAKENHI Grant Number JP24K00891. This work was supported by JST BOOST, Japan Grant Number JPMJBS2414. This work was supported by a grant from FUJIFILM Corporation. We note that you have provided funding information that is not currently declared in your Funding Statement. However, funding information should not appear in the Acknowledgments section or other areas of your manuscript. We will only publish funding information present in the Funding Statement section of the online submission form. Please remove any funding-related text from the manuscript and let us know how you would like to update your Funding Statement. Currently, your Funding Statement reads as follows:  This work was supported by JSPS KAKENHI Grant Number JP24K00891. This work was supported by JST BOOST, Japan Grant Number JPMJBS2414.  This work was supported by a grant from FUJIFILM Corporation.The funders had no role in study design, data collection and analysis, decision to publish, or preparation of the manuscript. Please include your amended statements within your cover letter; we will change the online submission form on your behalf. 5. Thank you for stating the following in the Competing Interests section: I have read the journal's policy and the authors of this manuscript have the following competing interests: SY and TS are employees of Fujifilm corporation.    We note that one or more of the authors are employed by a commercial company: Fujifilm Corporation.  a. Please provide an amended Funding Statement declaring this commercial affiliation, as well as a statement regarding the Role of Funders in your study. If the funding organization did not play a role in the study design, data collection and analysis, decision to publish, or preparation of the manuscript and only provided financial support in the form of authors' salaries and/or research materials, please review your statements relating to the author contributions, and ensure you have specifically and accurately indicated the role(s) that these authors had in your study. You can update author roles in the Author Contributions section of the online submission form. Please also include the following statement within your amended Funding Statement. “The funder provided support in the form of salaries for authors [insert relevant initials], but did not have any additional role in the study design, data collection and analysis, decision to publish, or preparation of the manuscript. The specific roles of these authors are articulated in the ‘author contributions’ section.”If your commercial affiliation did play a role in your study, please state and explain this role within your updated Funding Statement.  b. Please also provide an updated Competing Interests Statement declaring this commercial affiliation along with any other relevant declarations relating to employment, consultancy, patents, products in development, or marketed products, etc.   Within your Competing Interests Statement, please confirm that this commercial affiliation does not alter your adherence to all PLOS ONE policies on sharing data and materials by including the following statement: "This does not alter our adherence to  PLOS ONE policies on sharing data and materials.” (as detailed online in our guide for authors http://journals.plos.org/plosone/s/competing-interests) . If this adherence statement is not accurate and  there are restrictions on sharing of data and/or materials, please state these. Please note that we cannot proceed with consideration of your article until this information has been declared. Please include both an updated Funding Statement and Competing Interests Statement in your cover letter. We will change the online submission form on your behalf. 6. Please note that your Data Availability Statement is currently missing the repository name. If your manuscript is accepted for publication, you will be asked to provide these details on a very short timeline. We therefore suggest that you provide this information now, though we will not hold up the peer review process if you are unable. 7. When completing the data availability statement of the submission form, you indicated that you will make your data available on acceptance. We strongly recommend all authors decide on a data sharing plan before acceptance, as the process can be lengthy and hold up publication timelines. Please note that, though access restrictions are acceptable now, your entire data will need to be made freely accessible if your manuscript is accepted for publication. This policy applies to all data except where public deposition would breach compliance with the protocol approved by your research ethics board. If you are unable to adhere to our open data policy, please kindly revise your statement to explain your reasoning and we will seek the editor's input on an exemption. Please be assured that, once you have provided your new statement, the assessment of your exemption will not hold up the peer review process.

Reviewers' comments:

Reviewer's Responses to Questions

**Comments to the Author**

1. Is the manuscript technically sound, and do the data support the conclusions?

Reviewer #1: Yes

Reviewer #2: Yes

2. Has the statistical analysis been performed appropriately and rigorously? 

Reviewer #1: Yes

Reviewer #2: Yes

3. Have the authors made all data underlying the findings in their manuscript fully available?

Reviewer #1: Yes

Reviewer #2: Yes

4. Is the manuscript presented in an intelligible fashion and written in standard English?

Reviewer #1: Yes

Reviewer #2: No

5. Review Comments to the Author

**Reviewer #1: ** Reviewer recommendation: Accept with minor revisions.

Detailed review:

- Provide an abbreviation glossary and ensure each term is clearly defined when introduced.

- Include discussion points on how this work might influence clinical workflows and address any potential biases or risks in real-world applications.

- The reconstruction time (over 3 hours per volume for ZSSSL) is a significant limitation for clinical use. Expand the discussion of strategies to mitigate this issue, such as transfer learning or parameter reduction, in future work.

- Ensure figures and tables are fully interpretable with detailed captions and legible text.

- Strengthen the discussion around plans for prospective validation and explain how the method may perform under real-world imaging conditions.

**Reviewer #2:**  • The manuscript includes many technical terms and acronyms (e.g., CS, PI, SL, SSL, ZSSSL, MEL, GT, AF) without always defining them. This could make the text difficult to follow for readers outside the field.

• The manuscript is highly technical, and some sections (e.g., network architecture, training strategies) are difficult to follow for readers who are not experts in MRI reconstruction or deep learning.

• Add a background section to explain key terms, concepts, and MRI sequences. The paper would benefit from the addition of a background section to improve accessibility and readability. Many key concepts such as Self-Supervised Learning (SSL), Zero-Shot Self-Supervised Learning (ZSSSL), Memory Efficient Learning (MEL), and the qMRI sequences (MSME, VFA-SPGR, DESS) are central to the study but are not explained in sufficient detail for readers who may not be familiar with these techniques.

• The network architecture section, as currently written, is highly technical and assumes a deep familiarity with deep learning concepts and MRI reconstruction. Terms like "unrolled network," "conjugate gradient method," and "residual blocks" are used without sufficient explanation. The section does not explain why the specific network architecture was chosen or how it addresses the challenges of qMRI reconstruction.

• The mention of "Equation (S8)" and "Problem formulation" in the supplementary file (S1 File) is problematic because readers cannot access this information without switching to another document. Key equations and formulations should be included in the main text or explained in detail.

• The manuscript does not always explain why certain methods or parameters were chosen. For example, why were specific acceleration factors (AF4, AF8, AF12) selected? What is the rationale behind the 60%-40% split in SSL?

• The training does not explain why the specific training strategies were chosen or how they address the challenges of qMRI reconstruction.

• The training section mentions that "the lost computation results were recalculated from the cache during backpropagation," but it does not explain how this recalculation is performed or what its computational cost is. Provide more details about the implementation of MEL, such as how the cache is managed and how the recalculation process works.

• The presentation of quantitative results could be improved. Some tables and figures are difficult to interpret, and the text does not always provide clear explanations of the results.

• Some sentences are long and complex, which can make them harder to follow. simplify the language for broader readability

6. PLOS authors have the option to publish the peer review history of their article (what does this mean? ). If published, this will include your full peer review and any attached files.

**Do you want your identity to be public for this peer review?** For information about this choice, including consent withdrawal, please see our Privacy Policy .

Reviewer #1: **Yes: ** Hassan Mostafa Ahmed Hassan Fahmy

Reviewer #2: **Yes: ** Rasha Shoitan

---

## [Author Response · Author response to Decision Letter 1]

5 Apr 2025

This document (reply_to_reviewer.docx) contains the text of the response to the reviewer and the revised text of the manuscript. Here, the referee’s comments were marked in light blue, the responses were in black, and the revisions were recorded using the track changes in Word. The page and line numbers are those in the highlighted version of the revised manuscript (manuscript_rev_highlighted.docx).

In addition, to correct grammatical errors and improve overall readability, we used a professional English proofreading service by a native speaker. Please note that these corrections are not individually mentioned in the manuscript.

We have included three newly created figures (Fig7.tif, Fig9.tif, and Fig11.tif) to improve the visual presentation of the quantitative results. As a result of adding these figures, the numbering of the subsequent figures has shifted: the original Fig 7 is now Fig 8, Fig 8 is now Fig 10, and Fig 9 is now Fig 12 in the revised manuscript.

*Reviewer #1: *Reviewer recommendation: Accept with minor revisions.

Detailed review:

[R1-1] Provide an abbreviation glossary and ensure each term is clearly defined when introduced.

Thank you for pointing this out. In response to your comment, we have reviewed the manuscript to ensure that all abbreviations are clearly defined upon their first appearance. Specifically, we have revised the following sentences:

Introduction: page 3, line 2

Compressed sensing (CS)[7], which leverages the inherent sparsity of MR images and parallel imaging (PI)[8–11], which uses multiple receiver coils are also used to accelerate qMRI by reducing the number of data points acquired.

Introduction: page 3, line 6

The DL framework commonly used in image reconstruction is supervised learning (SL), where many fully sampled (FS) data that satisfy the Nyquist condition are used to train the model parameters for reconstruction from undersampled (US) data that do not satisfy the Nyquist condition. Reconstruction models using SL have also been applied to accelerate qMRI.

Material and Methods: page 8, line 22

The US k-space data used for training and testing were generated by retrospective sampling from the FS data under different acceleration factors (AFs), which represent the degree of acceleration relative to the acquisition time of the FS data (e.g., AF = 2 corresponds to a two-fold acceleration).

Additionally, to further improve clarity and address a similar concern raised by Reviewer 2 (please see [R2-3]), we have added a “Background” section. This new section introduces and explains the technical terms in this study, thereby enhancing the overall accessibility of the manuscript.

[R1-2] Include discussion points on how this work might influence clinical workflows and address any potential biases or risks in real-world applications.

Thank you for the comment. We have added a brief discussion on the clinical impact of our method, the need for standardization, and potential real-world biases. The revised text is as follows:

Discussion: page 19, line 20

The GT-free learning method and memory-efficient strategy used in this study are expected to shorten the total examination time of protocols incorporating conventional qMRI and enable the integration of qMRI sequences that were previously challenging due to long scan durations. SSL technology allows for the construction of DL models using only undersampled (US) data, even for qMRI sequences where acquiring fully sampled (FS) data is time-consuming. Additionally, ZSSSL facilitates the application of DL models to emerging qMRI sequences that are not yet widely adopted and lack large-scale training datasets. When incorporated into clinical workflows, these methods are expected to enhance objective diagnosis by utilizing qMRI as an imaging biomarker. However, to ensure the clinical relevance of observed quantitative values, a standardized workflow from image acquisition to quantitative analysis, including diagnosis and treatment planning, must be established. Such standardization will be essential alongside advancements in accelerated qMRI techniques.

[R1-3] The reconstruction time (over 3 hours per volume for ZSSSL) is a significant limitation for clinical use. Expand the discussion of strategies to mitigate this issue, such as transfer learning or parameter reduction, in future work.

Thank you for pointing out this important limitation. We have expanded the Discussion (page 20, line 17) to address potential strategies to reduce reconstruction time, such as transfer learning with SSL-pretrained weights, use of natural image or simulation-based pretraining, and model simplification via Neural Architecture Search. The revised text is as follows:

Material and Methods: page 8, line 22

Although ZSSSL is highly effective because it does not require a training dataset, it has the drawback of lengthy reconstruction time. In this study, the reconstruction time per volume exceeded three hours, which may be acceptable in preclinical settings but is not feasible for clinical use. One potential solution to reduce the reconstruction time is to employ transfer learning (TL) using weights pretrained with SSL. Yaman et al. [37] reported that this TL approach effectively improves performance and reduces reconstruction time to less than one-tenth. Alternatively, TL using weights pretrained with natural images [44,45] or simulation-based training datasets [46] may also be effective. Another approach is to decrease the number of model parameters, which can be achieved through Neural Architecture Search (NAS)[47]. NAS is an automated technique for finding optimal neural network architectures by exploring a predefined search space, potentially reducing the model size while maintaining or improving the performance. Because the network used in this study has not yet undergone sufficient tuning of the hyperparameters such as the number of filters and layers, optimizing the model architecture remains a topic for future work.

[R1-4] Ensure figures and tables are fully interpretable with detailed captions and legible text.

Thank you for your comment. We have revised the figure captions to include more detailed explanations and ensured that all abbreviations are defined. In addition, we have checked label clarity to improve readability. The updated captions can be found in the revised manuscript as follows:

Material and Methods: page 10, line 18

Fig 2. Architecture of the image reconstruction model used in this study. D_w corresponds to denoising in Equation (12) and DC corresponds to data consistency in Equation (13). The multi-contrast US k-space is the input and the multi-contrast reconstruction k-space is the output. Because MRI images are complex-valued, the input and output of the denoising unit are concatenated in real and imaginary parts in the channel dimension direction. US: Undersampled, CNN: Convolutional neural network.

Material and Methods: page 10, line 30

Fig 3. Training strategy in this study. The figure illustrates the training, validation, and testing procedures for supervised learning, self-supervised learning, and zero-shot SSL. Ω is the original US k-space. Θ, Λ, Ξ, and Γ are subsets of the k-spaces. US: Undersampled.

Material and Methods: page 14, line 27

Fig 4. Simulation procedures for Experiment 1 and Experiment 2. The blue box represents the simulation process of multi-contrast data. The yellow box illustrates the procedure for generating the fully sampled (FS) and undersampled (US) k-space data and coil sensitivity maps for training and testing in this study. US: Undersampled, FS: Fully sampled, GT: ground truth.

Material and Methods: page 14, line 31

Fig 5. coil arrangement used to simulate multi-coil signals. (Left) The red cube represents the field of view (FOV), showing the spatial arrangement of the receiver coils. Different colors indicate the wiring patterns of each coil. (Right) A 2D projection of the coil arrangement. Each coil is modeled as a loop coil with a radius of 100 mm.

Material and Methods: page 19, line 7

Fig 12. Memory consumption and training time per minibatch in each network parameter settings. (a) Maximum memory usage as a function of the number of filters, layers, and iterations. The solid blue line represents results with memory-efficient learning (w/ MEL), while the dashed orange line represents results without memory-efficient learning (w/o MEL). (b) Training time per minibatch.

[R1-5] Strengthen the discussion around plans for prospective validation and explain how the method may perform under real-world imaging conditions.

Thank you for the helpful comment. We have expanded the Discussion (page 21, line 22) to clarify the limitations of retrospective design and to describe plans for prospective validation. The revised text is as follows:

Discussion: page 21, line 22

Second, the design of this study was retrospective, and its performance has not been verified using prospective data. In actual scans, imperfect imaging conditions, such as inaccurate sampling patterns due to eddy current effects, may degrade the reconstruction performance. For example, the B0 magnetic field inhomogeneity and eddy currents can cause geometric distortion and signal loss [55]. Therefore, when planning a prospective study, it may be necessary, for example to assess the impact of eddy currents on image distortion and trajectory deviation through spatiotemporal measurement of k-space trajectories using a field camera [56], and to evaluate the distortion itself through phantom study [57]. Validation with prospectively acquired US k-space data is a future issue, particularly in a study design such as the present study, which used only US k-space data.

*Reviewer #2:*

[R2-1] The manuscript includes many technical terms and acronyms (e.g., CS, PI, SL, SSL, ZSSSL, MEL, GT, AF) without always defining them. This could make the text difficult to follow for readers outside the field.

Thank you for the comment. As noted in our response to [R1-1], we have revised the manuscript to define all technical terms and acronyms upon first use. In addition, we have added a Background section to improve clarity, as also explained in our response to [R2-3].

[R2-2] The manuscript is highly technical, and some sections (e.g., network architecture, training strategies) are difficult to follow for readers who are not experts in MRI reconstruction or deep learning.

Thank you for the comment. We have added explanations to improve clarity where possible. For specific revisions addressing this point, please refer to our responses to [R2-3], [R2-4], and [R2-8].

[R2-3] Add a background section to explain key terms, concepts, and MRI sequences. The paper would benefit from the addition of a background section to improve accessibility and readability. Many key concepts such as Self-Supervised Learning (SSL), Zero-Shot Self-Supervised Learning (ZSSSL), Memory Efficient Learning (MEL), and the qMRI sequences (MSME, VFA-SPGR, DESS) are central to the study but are not explained in sufficient detail for readers who may not be familiar with these techniques.

Thank you for the helpful suggestion. To improve accessibility and readability, we have added a new Background section. This section introduces key concepts, technical terms, and MRI sequences used in the study, including Self-Supervised Learning (SSL), Zero-Shot SSL (ZSSSL), and the qMRI sequences (MSME, VFA-SPGR, qDESS). The added section is located at the beginning of the manuscript and provides the necessary context for readers who may not be familiar with MRI reconstruction or deep learning. For a detailed explanation of Memory-Efficient Learning (MEL), please refer to our response to [R2-8].

Background

Problem formulation

In this section, we formulize the MRI sampling process. Under multi-coil conditions, the sampling process can be formalized as follows:

█(y_Ω=E_Ω x#(1) )

where x∈R^(H×W×2) is the target image, and y_Ω∈R^(C×H×W×2) represents the US multi-coil k-space data. Ω is the sampling region, C is the number of receiving coils, and H×W is the image matrix size. The encoding operator E_Ω:R^(H×W×2)↦R^(C×H×W×2) in PI is defined as:

█(E_Ω=[█(M_Ω FC_1@M_Ω FC_2@⋮@M_Ω FC_C )]#(2) )

where C_i is the sensitivity of the receiving coil, F is the discrete Fourier operator, and M_Ω is the sampling operator that fills unmeasured k-space points (outside Ω) with zeros.

In this study, we extended the encoding operator in Equation (2) to the sampling process for multicontrast data. In multicoil/multicontrast data acquisition, different sampling regions are often employed for each contrast; thus, the multicontrast extension of the encoding operator in Equation (2) is defined as:

█(E_Ω=[█(E_(Ω_1 )@E_(Ω_2 )@⋮@E_(Ω_P ) )]#(3) )

Thus, the operator E_Ω:R^(P×C×H×W×2)↦R^(P×H×W×2) is the encoding operator for multi-contrast k-space data, where P is the number of input contrast images, and Ω_i is the sampling region in k-space for the i-th contrast. Here, Ω={Ω_1,Ω_2,…,Ω_P} represents the sampling regions for all contrasts. Using this operator, the sampling operation for multi-coil / multi-contrast data can be expressed similarly to Equation (1). Henceforth, unless otherwise specified, the encoding operator in this paper refers to the multicontrast extended encoding operator defined in this section.

Quantitative MRI sequences in this study

This section provides an overview of the qMRI sequences used in the present study. In this study, MSME, VFA-SPGR, and qDESS were adopted as standard methods for 2D T2, 3D T1, and fast 3D T2 mapping, respectively.

MSME is a pulse sequence used for T2 mapping that combines multi-slice imaging that excites multiple slices in parallel with the spin-echo technique, where a 90-degree RF pulse excites nuclear magnetization and a 180-degree RF pulse refocuses it to generate echo signals. MSME acquires multiple spin echoes at different echo times (TE) from a single excitation, thus enabling efficient T2 mapping. The signal model is expressed as follows:

█(S_i (I_0,T_2 )=I_0 e^(-TE_i/T_2 )#(4) )

where I_0 and T_2 represent the proton density image and T2 map, respectively. 〖TE〗_i represents the echo time of the ith echo. Using the signal model in (4), the T2 map can be estimated using least-squares fitting (LSF).

VFA-SPGR is a widely used method for T1 mapping, particularly for acquiring 3D T1 maps [29]. This method uses SPGR sequences acquired multiple times with multiple FA and constant TR and TE. The signal model for the SPGR sequence is as follows:

█(S_i (I_0,T_1 )=I_0 ((1-e^(-TR/T_1 )))/(1-e^(-TR/T_1 ) cosα_i )#(5) )

where α_i is the flip angle of the ith measurement. If there are at least two measurements with different flip angles, T1 mapping is possible using Equation (5).

qDESS is a 3D T2 mapping method based on the DESS sequence, a well-known, undistorted SNR-efficient 3D imaging technique [30]. In DESS, two contrast images, free induction decay (FID) signals, S_fid, and ECHO signals, S_echo, are obtained simultaneously per TR. T2 mapping is performed by utilizing the relationship between the FID signal and ECHO signal ratio as follows [31]:

█(S_echo/S_fid =e^(-2(TR-TE)/T_2 )#(6) )

In a study by Sveinsson et al. [32], T1 and diffusion effects were considered, particularly when focusing on the knee.

█(S_echo/S_fid =e^(-2(TR-TE)/T_2-(TR-τ/3)Δk^2 D ) sin^2⁡〖(α/2)〗 ((1+e^(-TR/T_1-TRΔk^2 D))/(1-cosαe^(-TR/T_1-TRΔk^2 D) ))#(7) )

where TR and TE represent the repetition and echo time, respectively. α is the flip angle, D is the diffusivity, and the dephasing per unit length induced by the unbalanced gradient is denoted by Δk =γGτ, where G and τ are the spoiler amplitude and duration, respectively. In this study, following Sveinsson et al.[32], we focused on knee imaging and assumed T1 to be 1000 ms, with other sequence parameters based on the SKM-TEA dataset [33].

Training strategy of deep learning MRI reconstruction

In this section, the DL training method used in this study is described. In recent years, research on learning methods has been actively conducted with the development of DL networks. The datasets required differed depending on the learning method, and some datasets did not require a ground truth (GT). The represe

---

## [Decision Letter · Decision Letter 1]

28 Apr 2025

Ground-truth-free deep learning approach for accelerated quantitative parameter mapping with memory efficient learning

PONE-D-24-54559R1

Dear Dr. Terada,

We’re pleased to inform you that your manuscript has been judged scientifically suitable for publication and will be formally accepted for publication once it meets all outstanding technical requirements.

Kind regards,

Ibrahim Sadek, Ph.D.

Academic Editor

PLOS ONE

Additional Editor Comments (optional):

Reviewers' comments:

Reviewer's Responses to Questions

**Comments to the Author**

1. If the authors have adequately addressed your comments raised in a previous round of review and you feel that this manuscript is now acceptable for publication, you may indicate that here to bypass the “Comments to the Author” section, enter your conflict of interest statement in the “Confidential to Editor” section, and submit your "Accept" recommendation.

Reviewer #1: All comments have been addressed

Reviewer #2: All comments have been addressed

2. Is the manuscript technically sound, and do the data support the conclusions?

Reviewer #1: Yes

Reviewer #2: Yes

3. Has the statistical analysis been performed appropriately and rigorously? 

Reviewer #1: Yes

Reviewer #2: Yes

4. Have the authors made all data underlying the findings in their manuscript fully available?

Reviewer #1: Yes

Reviewer #2: (No Response)

5. Is the manuscript presented in an intelligible fashion and written in standard English?

Reviewer #1: Yes

Reviewer #2: No

6. Review Comments to the Author

Reviewer #1: There are no further comments. The authors have addressed properly the comments that are sent to them.

Reviewer #2: Please improve the resolution of the figures to ensure clear visualization.

Please consider revising the manuscript for improved clarity and smoother flow, as some sections are difficult to follow and may benefit from enhanced organization

7. PLOS authors have the option to publish the peer review history of their article (what does this mean? ). If published, this will include your full peer review and any attached files.

**Do you want your identity to be public for this peer review?** For information about this choice, including consent withdrawal, please see our Privacy Policy .

Reviewer #1: **Yes: ** Hassan Mostafa Ahmed Hassan Fahmy

Reviewer #2: No

---

## [Editor Report · Acceptance letter]

PONE-D-24-54559R1

PLOS ONE

Dear Dr. Terada,

I'm pleased to inform you that your manuscript has been deemed suitable for publication in PLOS ONE. Congratulations! Your manuscript is now being handed over to our production team.

Kind regards,

on behalf of

Dr. Ibrahim Sadek

Academic Editor

PLOS ONE